# The organization of double-stranded RNA in the chikungunya virus replication organelle

Timothée Laurent [1,2,3,4], Lars-Anders Carlson [1,2,3,4] *

**1** Department of Medical Biochemistry and Biophysics, Umeå University, Umeå, Sweden, **2** Wallenberg Centre for Molecular Medicine, Umeå University, Umeå, Sweden, **3** Molecular Infection Medicine Sweden, Umeå University, Umeå, Sweden, **4** Umeå Centre for Microbial Research (UCMR), Umeå, Sweden

* lars-anders.carlson@umu.se

**Data Availability Statement:** The subtomogram averages of dsRNA have been deposited at the electron microscopy data bank (EMDB) with the accession codes EMD-16283 (class 1), EMD-16298 (class 2), EMD-16300 (class 3), EMD-

## Abstract

Alphaviruses are mosquito-borne, positive-sense single-stranded RNA viruses. Amongst the alphaviruses, chikungunya virus is notable as a large source of human illness, especially in tropical and subtropical regions. When they invade a cell, alphaviruses generate dedicated organelles for viral genome replication, so-called spherules. Spherules form as outward-facing buds at the plasma membrane, and it has recently been shown that the thin membrane neck that connects this membrane bud with the cytoplasm is guarded by a two-megadalton protein complex that contains all the enzymatic functions necessary for RNA replication. The lumen of the spherules contains a single copy of the negative-strand template RNA, present in a duplex with newly synthesized positive-sense RNA. Less is known about the organization of this double-stranded RNA as compared to the protein components of the spherule. Here, we analyzed cryo-electron tomograms of chikungunya virus spherules in terms of the organization of the double-stranded RNA replication intermediate. We find that the double-stranded RNA has a shortened apparent persistence length as compared to unconstrained double-stranded RNA. Around half of the genome is present in either of five conformations identified by subtomogram classification, each representing a relatively straight segment of ~25–32 nm. Finally, the RNA occupies the spherule lumen at a homogeneous density, but has a preferred orientation to be perpendicular to a vector pointing from the membrane neck towards the spherule center. Taken together, this analysis lays another piece of the puzzle of the highly coordinated alphavirus genome replication.

## Author summary

Alphaviruses, such as chikungunya virus, are mosquito-borne viruses that cause various diseases in humans and other animals. When they infect a cell, they hijack the plasma membrane to generate dedicated genome-replicating organelles called spherules. Spherules contain the negative strand "template" to make more positive-strand viral RNA. The template is present in a double-stranded duplex form (dsRNA) with a positive strand RNA. Here, we used cryo-electron tomography and computational methods to study how the dsRNA organizes inside spherules. We find that the dsRNA is more strongly bent than

16301 (class 4) and EMD-16302 (class 5). The two tomograms analyzed in the present publication are available at the electron microscopy databank (EMDB) with accession numbers EMD-15582 and EMD-15583.

**Funding:** This project was funded by the Swedish research council ("Vetenskapsrådet", grants 2018-05851, 2021-01145) and a Human Frontier Science Program Career Development Award (CDA00047/2017-C) to L.-A.C. The funders had no role in study design, data collection and analysis, decision to publish, or preparation of the manuscript.

**Competing interests:** The authors have declared that no competing interests exist.

it would be outside a spherule. This is measured by the so-called "persistence length", which can be seen as the distance over which a polymer is roughly straight. For free dsRNA the literature reports this as 63 nm, but we measure 22 nm in spherules. We corroborated this by a method called subtomogram averaging which reveals a handful prevalent conformations of dsRNA in spherules. We further showed that dsRNA is evenly distributed inside the spherule, and has a certain degree of preferred orientation with respect to the protein machinery that replicates it, the so-called neck complex.

## Introduction

The chikungunya virus (CHIKV) is a mosquito-borne virus belonging to the *Alphavirus* genus in the family *Togaviridae*. It is responsible for the emerging disease Chikungunya fever, characterized by fever, rashes and a crippling arthralgia with symptoms that may last up to several months or even years [1]. The virus is transmitted through mosquito bites of *Aedes aegypti* and *Aedes albopictus* that are found in South America, Africa, Southeast Asia and in the most recent years, as north as the United States and France [2]. A large number of alphaviruses infect mammals. Some (e.g., Sindbis virus, Semliki forest virus, O'Nyong Nyong virus) cause chikungunya-like diseases in humans, whereas another group of alphaviruses, known as New-World alphaviruses, cause serious encephalitic conditions in e.g. horses [3, 4]. The potential for further spread of disease-causing alphaviruses is highlighted by the plethora of alphaviruses that infect mammals, and the well-described event in 2006 where CHIKV mutated such that it could be spread by a different mosquito species, *Aedes albopictus* [5]. The changing habitats of the mosquitoes due to climate change provide an additional reason to monitor emerging alphaviruses [6].

Alphaviruses are positive-sense single-stranded RNA (+ssRNA) viruses. They follow the general principle of such viruses to rearrange host-cell cytoplasmic membranes and turn them into RNA replication organelles [7–9]. In the case of the *Alphaviruses*, these organelles are referred to as "spherules" [10]. Spherules are ~60–80 nm bud-shaped membrane evaginations that initially form at the plasma membrane, after which they—for some alphaviruses–are trafficked to modified endosomes, referred to as cytopathic vesicles [11]. The spherules serve the purpose of replicating the viral RNA while hiding the double-stranded RNA (dsRNA) replication intermediate from innate immunity detection [12].

The alphavirus genome is composed of two open reading frames (ORFs): the first one codes for non-structural proteins, i.e. viral proteins required for replication of the viral genome, and a second one codes for structural proteins, which constitute viral particles. These two ORFs are translated from two separate RNA molecules–the genomic and sub-genomic RNA–both of which are produced by the spherule from the same, full-length template RNA of negative polarity [13]. Each spherule contains a single copy of this negative strand template RNA, and it is to ~80% present in a double-stranded duplex with positive-strand RNA [14]. The thin membrane neck that separates the spherule lumen from the cytoplasm is guarded by a two-megadalton protein complex [14, 15]. The base of this "neck complex" is the membrane-associated viral capping enzyme nsP1. NsP1 further positions the viral helicase (nsP2) and RNA-dependent RNA polymerase (nsP4) at the center of the neck [15]. This neck complex structure suggested a replication mechanism where the template strand (in the form of dsRNA) is continuously threaded through the polymerase at the luminal side of the neck complex, resulting in extrusion of the displaced positive-sense strand through the neck complex into the cytoplasm.

Spatially unconstrained dsRNA has been extensively characterized and it is known that the double helix adopts A-form [16], has a diameter of 24 Å [17] and a persistence length of 63 nm [18, 19]. The persistence length of semi-rigid polymers, such as nucleic acids, is a measurement of its bending rigidity. In qualitative terms it can be described this way: If the length (L) of the polymer is smaller than its persistence length (P), it will behave as a rod-like structure. If L is greater than P, the polymer will appear bendable. Quantitatively, the persistence length is calculated from the mean value of the cosines between the angles **θ** formed by the tangents at two point x and x+X along the length of the filaments. This translates into ‹cos**θ**(X)›~exp(-X/P). Thus, the persistence length can be estimated by plotting ln(‹cos**θ**(X)›) as a function of X, and finding P on the resulting graph as the value at which the extrapolation of the linear segment of the curve crosses the X-axis [20].

Less is known about the behavior of dsRNA in spatial confinement such as a spherule. At the extreme end of the spectrum, it has been shown that virions with a dsRNA genome organize their genetic material in highly ordered structures in their capsids [21–23]. Having deformable membranes and being subject to constant reorganization of the dsRNA due to the action of the polymerase, this scenario appears less likely for spherules. However, the dsRNA contents of spherules is still unlikely to be unaffected by their confinement, alone for the reason that the persistence length of dsRNA roughly equals the diameter of the spherules. Theoretical treatments of biological filaments are often based on the theory of worm-like chains. This theory describes the behavior of linear semi-rigid cylindrical polymers in terms of e.g. persistence length. The worm-like chain model has been shown to be suited to describe dsDNA [24, 25]. Applying this theory to biological filaments in confinement have shown that the arrangement of the filaments depends on the packaging density, leading to an isotropic occupancy at lower density of filaments or to three possible ordered liquid-crystalline spooling states at higher densities [26]. Furthermore, it has been shown that under spherical confinement, the persistence length of a polymer appears to be shorter, which is referred to as "apparent persistence length" [27].

Little is known about the organization and physical properties of dsRNA within alphavirus spherules. Since their interior is separated from the cytoplasm of the host cell, it is possible that the solvent differs from the environment in the cytoplasm, which may affect the properties of the dsRNA. To investigate the organization of dsRNA inside spherules, we analyzed cryo-electron tomograms of CHIKV spherules. We calculated subtomogram averages of different conformations taken by the dsRNA within spherules, and further analyzed its properties in terms of apparent persistence length, occupancy and preferred orientation with respect to the neck complex.

## Material and methods

### Sample preparation and cryo-electron tomography

The acquisition of all data analyzed in this publication was detailed in our previous study [14]. In brief, CHIKV viral replicon particles (VRPs) were produced by electroporation of three RNAs into baby hamster kidney (BHK) cells [28]. The VRPs were used to transduce BHK cells grown on EM-grids 6h prior to plunge-freezing in a liquid ethane/propane mix [14]. Tilt series for cryo-electron tomography were acquired at a 4.36 Å/px object pixel size. The two tomograms analyzed in the present publication are available at the electron microscopy databank (EMDB) with accession numbers EMD-15582 and EMD-15583.

### RNA tracing

Two high quality tomograms were binned four times and imported into the Amira software (Thermo Fisher Scientific) where the RNA tracing was performed using its filament tracing

functionality [29]. A light Gaussian filter was applied to the tomograms to improve the visibility of the filaments contained in the spherules. A cylinder correlation was applied on the tomogram to match the dsRNA. In short the settings were: cylinder length: 6 nm, angular sampling: 5˚, Mask cylinder radius: 5 nm, Outer cylinder radius: 3.5 nm, inner cylinder radius: 0 nm. The missing wedge correction function was applied. The interior of spherules was manually segmented to define the search region. Correlation lines were then traced using the following parameters: minimum seed correlation: 65, minimum continuation: 45, direction coefficient: 1, minimum length: 20 nm, minimum distance: 5 nm, angle: 180˚, minimum step size: 5%. In total, 79.3% of the total theoretical length of the dsRNA could be traced in 22 spherules.

## Subtomogram averaging

Subtomograms containing dsRNA were picked from two high quality tomograms binned by a factor two (voxel size 8.58 Å) using published scripts that reformat the coordinates from Amira to the subtomogram averaging software Dynamo [30–32]. To extract the subtomograms, a cropbox of 50 binned voxels was used and particles were picked every 20 binned voxels. The extraction yielded 905 particles. Those particles were used to generate an initial average, with an initial alignment based on the filament tracing (S1 Fig). Next, we randomized the azimuthal angles to decrease the impact of the missing wedge on the structure. A cylinder mask was applied and a multi-reference alignment was performed, for which five initial references were calculated from a small number of randomly selected subtomograms. During alignment, shifts were limited to 6 pixels in order to not allow the average to align to adjacent membrane or dsRNA densities. Filaments were allowed to tilt by 45˚ and a full azimuthal rotation was permitted. The resolution of each category was then estimated using the Gold-Standard Fourier Shell Correlation with a cut-off of 0.143 and each average were then low-pass filtered to their respective resolution. Structures were then segmented in UCSF Chimera [33].

## Relative orientation of the dsRNA with respect to the neck complex

Using MATLAB (Mathworks, Inc), the traced dsRNA was vectorized by computing vectors between the nodes used for the tracing. For each spherule, a reference vector pointing through the neck complex towards the center of the spherule was manually defined. For the 22 spherules, the angles between the dsRNA vectors and the reference vector were calculated and plotted.

## Estimation of the density of the RNA within spherules

The centre of gravity of each spherule were computed using the coordinates used for the tracing. The distance of each coordinate from the center of gravity was then calculated and the density was estimated by taking into account the volume of the spherule at the specific radial distance of each coordinate.

## Calculation of the volume fraction of the dsRNA within single spherules

The dsRNA was assumed to be a cylinder with a radius of 12 Å and its volume was calculated based on the length of the traced RNA in individual spherules. The volume fraction was then estimated by dividing the volume of the cylinder by the total volume of the corresponding spherule, as measured using volume segmentation in Amira (Thermo Fisher Scientific).

### Estimation of the concentration of polymerized NTPs within spherules

The number of moles of NTPs in each traced dsRNA was calculated and the concentration was further estimated by dividing this number by the volume of the spherule in which the molarity was calculated. The average molecular weight of a monophosphate ribonucleotide (339.5g/mol) was used to estimate the concentration of dsRNA in milligram per milliliter.

### Persistence length

The apparent persistence length was calculated using previously published scripts [34] based on the decay of the correlation between the cosines of the angles of the tangents calculated along the dsRNA.

## Results

### Subtomogram averaging unveils different conformation of the dsRNA in chikungunya spherules

We previously showed that cryo-electron tomography can resolve structural details of the CHIKV spherule [14]. A first characterization of the filamentous material in the spherule lumen indicated that each spherule contains a single copy of the viral genome, ~80% of which is present in dsRNA form. In this study, we aimed to further characterize the dsRNA packaging in spherules. An automated filament tracing algorithm that was developed for cryo-electron tomograms [29], and previously applied to e.g. filamentous actin [29, 35, 36] was used to trace the dsRNA contents of spherules (Fig 1A and S1 Video). This approach has been reported previously and has been shown to be quantitative and the output could also be used to structurally characterize the traced filaments [14, 29, 34, 37]. The tracing produced sets of 3D coordinates of enumerated points that define one filament. Typically, the tracing was not able to connect all filamentous density in one spherule lumen to a single filament, probably due to limited signal-to-noise ratio or interspersed ssRNA regions. However, several long continuous filaments were traced and the median filament length was 89 nm (S2 Fig). Since the reported 63 nm persistence length of dsRNA is very similar to the spherule diameter, dsRNA confined in spherules would be expected to have a shorter apparent persistence length [18, 27]. In agreement with this assumption, the estimated apparent persistence length in 12 individual spherules was on average 22.3 ± 3.8nm (Figs 1B and S3).

To characterize whether there are preferred conformations of the dsRNA in spherules, we next extracted subtomograms along the traced coordinates and subjected them to alignment, multireference classification, and averaging [38]. Subtomograms were extracted from 22 spherules and classified into five different classes without use of any external reference structure. All of the resulting classes showed long dsRNA stretches (~25–30 nm) in distinct conformations and different surroundings (Fig 1C–1G and Table 1). Strikingly, Class 1 showed clear density for the spherule membrane, and a dsRNA filament that closely followed the curvature of the membrane (Fig 1C). The distance between the dsRNA and the inner leaflet of the spherule membrane was ~25 Å. Classes 2–3 also showed dsRNA which was curved, with peripheral densities possibly corresponding to adjacent dsRNA or membrane (Fig 1D–1E). Further classes showed straighter sections of dsRNA, with variable adjacent densities (Fig 1F–1G). We measured the lengths of the continuous dsRNA densities in each class (Fig 1C–1G, red densities and Table 1). Knowing how many particles each class consisted of, we could thus calculate the total genome length present in that class. This can be expressed as the length per spherule, in base pairs of dsRNA, and hence as a percentage of the 8820 bp chikungunya replicon genome (Table 1 and Fig 1H). This showed that 52.4% of the CHIKV genome was present in

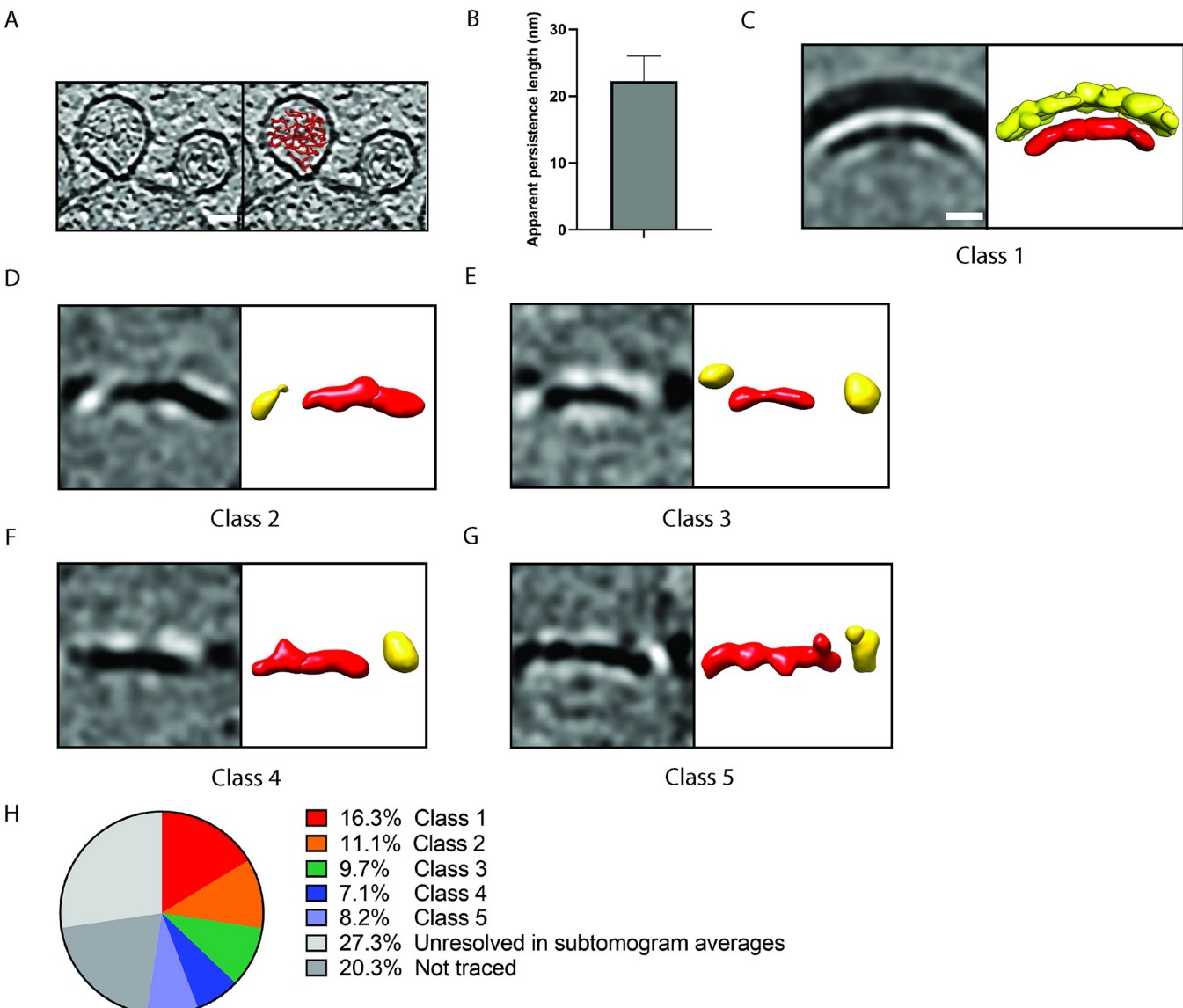

**Fig 1. Organization of the dsRNA in CHIKV spherules.** (A) Computational slice through a cryo-electron tomogram containing two CHIKV spherules at the plasma membrane. The right slice shows an overlay of the dsRNA tracing for one of the two spherules. Density is black. Scale bar, 25nm. (B) Average apparent persistence length of the dsRNA, individually measured in 12 spherules, with standard deviation indicated. The measurement was made using cosine correlation plots, one example of which is shown in S3C–S3G Fig. Unmasked subtomogram averages of five classes of dsRNA resulting from multireference alignment classification. For each class, the left panel shows a slice through the density. The right panel shows an isosurface representation where the density interpreted as dsRNA is shown in red, and adjacent densities in yellow. Scale bar, 10 nm. (H) Chart of the distribution of the genome percentage present in each average. The percentages of the dsRNA that was not present in our initial tracing, or traced but not resolved during the averaging process are also indicated.

well-ordered parts of the subtomogram averages. Another 27.3% was included in the filament tracing but not seen in the well-ordered parts of the subtomogram averages, and a further 20.3% were not identified in the filament tracing (Table 1 and Fig 1H).

Taken together, our data show that the dsRNA form of the genome contained within alphavirus spherules has a shortened apparent persistence length, but that >50% of the genome is present in stretches or relatively straight dsRNA of ~25–30 nm length.

**Table 1. Distribution of subtomograms in the five different classes shown in Fig 1.** The length of the dsRNA densities as measured in the averages is stated, as well as their contribution to the total length of the genome. The sum indicates the total percentage of the genome that could be resolved in the subtomogram average classes. This number (52%) is smaller than the fraction of the genome initially traced (80%) since only the consensus structure appears in the subtomogram average. Number of particles refers to the total number of subtomograms, from 22 spherules.

| Class | Length (nm) | Number of particles | Total length per spherule (bp) | % genome |
| --- | --- | --- | --- | --- |
| 1 | 32.3 | 171 | 1432 | 16.3 |
| 2 | 24.5 | 143 | 980 | 7.1 |
| 3 | 24.9 | 192 | 622 | 9.7 |
| 4 | 30 | 135 | 849 | 8.2 |
| 5 | 30.5 | 264 | 723 | 11.1 |
| Sum | 142.2 | 905 | 4606 | 52.4 |

## The RNA occupies the spherule lumen uniformly

So far, we investigated the conformation of sections of the dsRNA within spherules, but did not yet study its overall packaging. Using the volume measurements we previously performed for individual spherules [14], and assuming dsRNA to have a diameter of 24 Å (excluding its hydration shell), we found that the traced amount of dsRNA occupied 7.6 ± 0.9% of the spherule lumen (Fig 2A). This can also be described in terms of the density of genetic material in the spherules, i.e. the concentration of polymerized ribonucleotides which was found to be 0.22 ± 0.02 M, equivalent to 148 ± 17 mg/mL of dsRNA (Fig 2B).

Theoretical studies of genome arrangement in virus capsids have suggested that different average densities of the genetic material may lead to it either being homogeneously or inhomogeneously distributed in the confined space [26]. To investigate this in spherules, we used the traced coordinates to measure the distance of each filament point in a spherule from the center of the spherule. This revealed that the dsRNA uniformly occupies the lumen of the spherule (Fig 2C). The radius of the individual spherules used for this measurement varied between 30 and 38 nm. Hence the decreasing density of the dsRNA at the outer edge of the distribution reflects the heterogeneity of the radius of spherules. The uniform dsRNA density in the spherules is in line with the subtomogram classification that suggested that most dsRNA has no strong membrane density adjacent to it (Fig 1C–1H). Taken together, the dsRNA contents of spherules appears to have a homogeneous density in the spherical lumen, as consistent with a lower-density regime in theoretical studies [26].

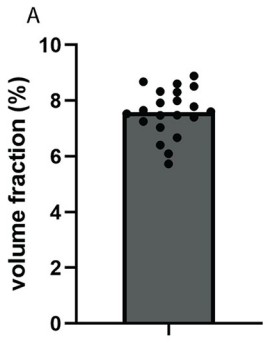
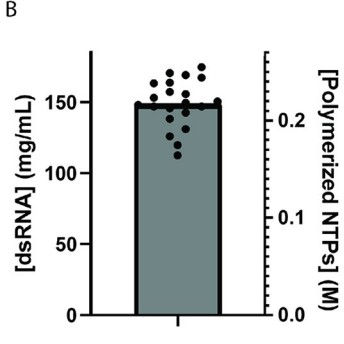
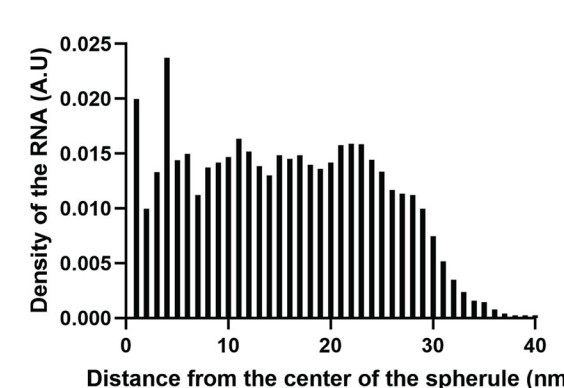

**Fig 2. Occupancy of the dsRNA within spherules.** (A) Volume fraction of the dsRNA, assuming a radius of 1.2 nm and using individually measured spherule volumes measured in tomograms. Each dot represents one spherule and the bar shows the mean. (B) Concentration of dsRNA in the spherules, expressed as molar concentration of polymerized NTPs and mg/ml dsRNA. Each dot represents one spherule and the bar shows the mean. (C) Average local density of dsRNA in 22 spherules as a function of the radial distance from the center of the spherule. The included spherules had radii in the range of 30–38 nm.

## dsRNA has a moderate preferential orientation within spherules

In virions, genomic nucleic acid is sometimes found in a single, defined conformation [21–23]. But spherules, as RNA replication machines, repeatedly pass the entire genome through the protein complex at their membrane neck where the RNA-dependent RNA polymerase is located [15]. Elsewhere in the spherule, the genome is likely to be relatively unconstrained except for the spatial confinement. We hypothesized that the replication machinery at the spherule neck might still generate some preferred orientation of the dsRNA in the spherule. To test this, we proceeded to measure the relative orientation of the traced dsRNA with respect to a vector parallel to the neck complex, pointing through the membrane neck towards the spherule center (Fig 3A). Three different principal scenarios for the orientation of the dsRNA within spherules would result in three different distributions of orientations. In the first, the

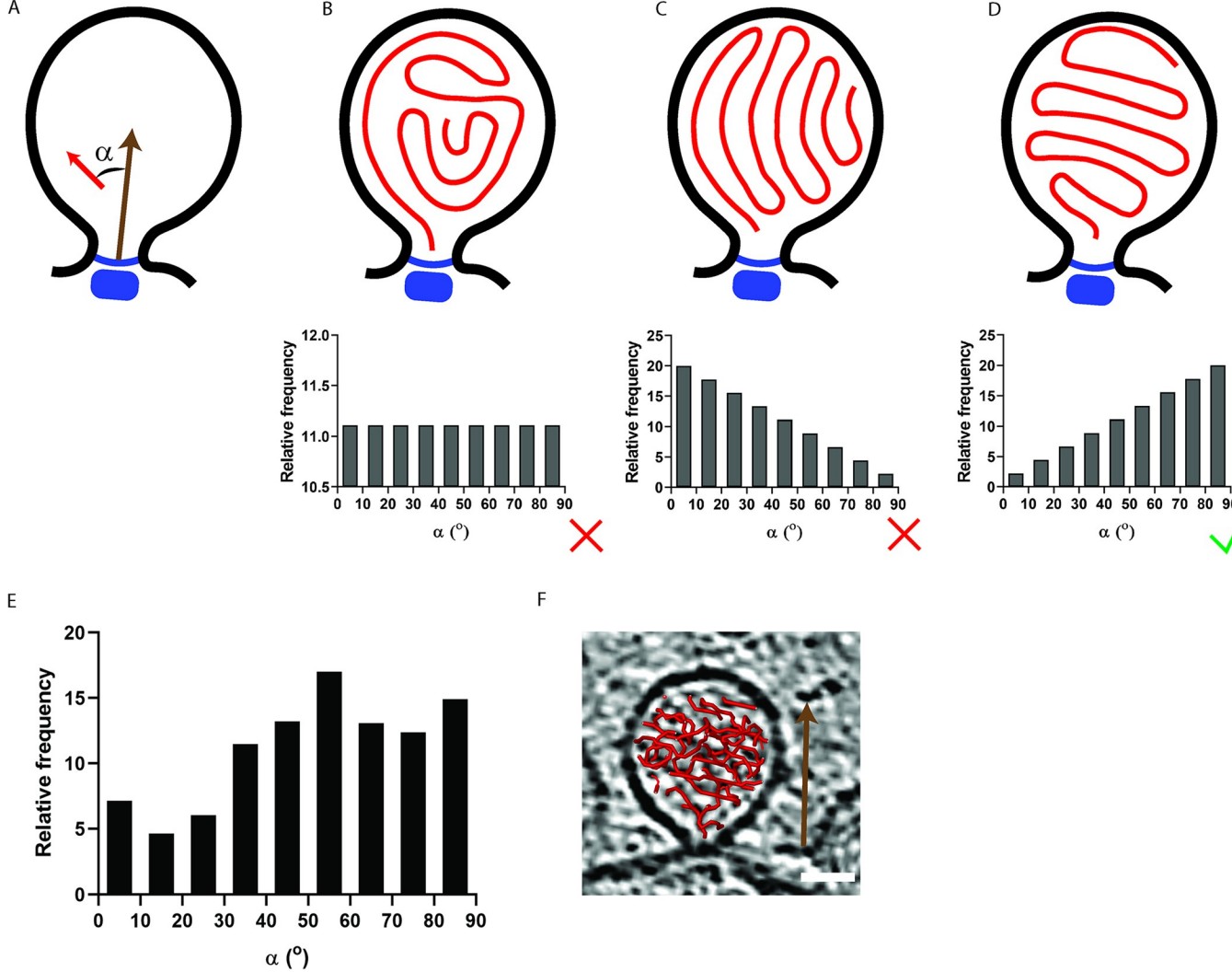

**Fig 3. Preferential orientation of the dsRNA in spherules.** (A) Schematic of the measurement of the angles between short segments of traced dsRNA and a reference vector anchored to the neck complex and pointing toward the spherule center. (B-D) Hypothetical relative frequency of angles, as defined in A, for the scenarios indicated in the spherule sketches. The dsRNA could pack with no preferred orientation to the neck complex vector (e.g., randomly) (B), with a preferred direction parallel (C), or perpendicular (D) to the neck complex vector. (E) Measurement of the relative frequency of angles in 22 spherules. (F) The traced dsRNA overlayed on a slice through the tomogram for one representative spherule included in the analysis. Scale bar, 25 nm.

dsRNA has no preferred orientation with respect to the neck complex. This could result from a random orientation of dsRNA sections, or from other arrangements, but they would share the feature that all measured angles would have nearly the same probabilities (Fig 3B). In two other scenarios, the dsRNA would either preferentially align along the axis of the neck complex (Fig 3C), or perpendicular to it (Fig 3D). We conducted this measurement on the 22 spherules for which dsRNA had been traced. The 22 spherules had neck complex vectors that were mostly near-parallel to the XY plane of the tomograms, but they were rotated differently. This means that the anisotropic resolution due to the missing wedge effect is unlikely to have introduced a bias in the measurements [39]. The resulting distribution is consistent with the third scenario that models the orientation of the dsRNA as preferentially being orthogonal to the neck complex vector (Fig 3E). A representative spherule illustrating this situation is shown in Fig 3F. Hence, while we do not expect a single global conformation of the dsRNA in alphavirus spherules, we can detect a preferred orientation of the dsRNA with respect to the neck complex.

## Discussion

All life forms rely on replicating and storing genetic information at a high spatial density. As cell-dependent replicators, viruses also adhere to this principle. The extreme case for viruses are their extracellular manifestations, the virions. Whether their genetic material is single or double-stranded, RNA or DNA, virions have frequently been shown to keep the entire viral genome in a single, precise spatial conformation inside the capsid [21–23]. In this study, we instead sought insights into the organization of the dsRNA in RNA virus replication organelles, the CHIKV spherules. Considering the nature of the spherules, there are *prima facie* several reasons to believe that the genome would be less organized than in virions: First, the template RNA is constantly passed through a single polymerase molecule which remains fixed at the membrane neck, hence a single placing of the genome can be excluded [15]. Second, the spherule is not delimited by a rigid protein capsid, but by a bud of the cell's plasma membrane which adjusts its size in response to the pressure exerted by the confined dsRNA [14]. Third, according to current understanding there are no specific interactions between this membrane and the confined dsRNA. In cryo-electron tomograms of CHIKV spherules, we estimated the apparent persistence length of the dsRNA to ~22 nm (Fig 1B). This is roughly 1/3 of the persistence length reported for dsRNA in a non-confined environment (63 nm), meaning that the dsRNA inside a spherule is substantially more bent than it would be in a free solution [18, 19]. The tracing of dsRNA in cryo-electron tomograms will by necessity be affected by noise, which will reduce the estimate for the apparent persistence length. Thus, our value of 22 nm should be considered as a lower bound for the apparent persistence length of dsRNA in spherules. It would however be impossible for the dsRNA to fit in a spherule of 60–80 nm diameter unless its apparent persistence length were shorter than 63 nm, and the fact that the apparent persistence length in spherical confinement is smaller than its non-confined counterpart is in accordance with the literature [40]. It is possible that other factors that the confinement also decrease the apparent persistence length of dsRNA in spherules: the RNA may be associated with proteins that aid its bending (three of four non-structural proteins of alphaviruses bind RNA), and interspersed single-stranded RNA regions may increase its bending since ssRNA has a persistence length of only ~1 nm [19]. The classification of subtomograms was consistent with the global analysis of apparent persistence length, since the lengths of the dsRNA densities observed in all five classes, at 25–32 nm, were close to the measured apparent persistence length (Table 1). The total resolved amount of dsRNA in the five classes reflects 52% of the genome length. The remaining 48% of genome might be present in more variable dsRNA conformations that were averaged out, or as single-stranded RNA [14].

We found that the dsRNA occupies the spherule lumen at a homogeneous concentration corresponding to 148 mg/ml, or a volume fraction of ~8%. The lack of concentration inhomogeneities stands in contrast to observations of liquid-crystal-like layers of different dsRNA concentration in virus capsids [21]. This can be discussed in the light of a theoretical study that treated genome organization in capsids using a mathematical model of worm-like chains in confinement [41]. In that study, it was found that two parameters define a phase diagram that dictate whether the genome assumes an isotropic, homogeneous distribution, or various inhomogeneous, liquid-crystal-like arrangements. The first parameter is the degree of confinement P/R, where P is the persistence length and R is the radius of the sphere in which it is confined. Assuming that spherules are spheres of 32 nm radius, and using the literature value of P = 63 nm for dsRNA, we arrive at P/R = 1.97 which is in the regime of strong confinement P/R >1. The second parameter is the "average relative rod density" $\rho_0$:

$$\rho_0 = \frac{3}{2\pi} * \frac{d}{P} * \left(\frac{P}{R}\right)^3 * \frac{L}{P}$$

where d is the diameter of the RNA (2.4 nm), and L the total length of the dsRNA (2253 nm for one replicon genome). For the dsRNA in spherules this gives $\rho_0$ = 4.96. In the phase diagram sketched by Liang *et al*, a confinement of 1.97 and $\rho_0$ of 4.96 is firmly in the isotropic phase, concurrent with our measurements [26]. In this respect, the theory of worm-like chains in confinement thus accurately describes dsRNA in spherules. In fact, at P/R = 2, a $\rho_0$ of >~14 would be required to enter the liquid crystal regime of non-isotropic distribution. This would correspond to roughly 2.8 times the amount of dsRNA present in the same spherule volume. Hypothetically, if the true persistence length of dsRNA were in fact closer to our measured $P_{apparent}$ (22.3 nm) rather than the published 63 nm, we would obtain P/R = 0.72, which is a weak confinement. The theory would still predict isotropic distribution.

Finally, we observed that the dsRNA has a moderately preferred orientation in spherules and tends to align itself orthogonally to a reference vector pointing through the spherule neck into its lumen (Fig 3). This observation diverges from the behavior expected in the isotropic distribution regime described above. We suggest that this deviation stems from the fact the spherules are dynamic structures where RNA synthesis constantly takes place at a single site (the neck complex).

Taken together, our analysis provides first insights into the spatial organization of the double-stranded RNA replicative intermediate of the alphavirus replication organelles. Our analysis is limited by the currently attainable resolution of cryo-electron tomograms, which precludes more detailed analyzes of RNA conformations, e.g. visualization of single-stranded regions. In future research, especially with improved tomogram resolution, it may be possible to e.g. correlate the observed conformations of RNA with their role in the genome replication. This methodology could also be applied to replication organelles formed by other +ssRNA viruses. It may highlight similarities and discrepancies during the replication of the viral genome and could potentially help to characterize the replication mechanism.

## Supporting information

**S1 Fig. dsRNA subtomogram processing.** Workflow used in subtomogram alignment, classification and averaging to compute the five averages of dsRNA shown in Fig 1C–1G. (TIF)

**S2 Fig. Length of traced dsRNA fragments.** The histogram shows the distribution of lengths of uninterrupted filament fragments traced in spherules. The median value is 88.8 nm,

corresponding to 347 base pairs.
(TIF)

**S3 Fig. Cosine correlation plot to estimate apparent persistence length of dsRNA.** The outcome for one representative spherule of the estimation of the apparent persistence length of the dsRNA using the correlation decay between the cosines of tangents. Blue dots: linear portion of the curve. Red crosses: correlation values beyond the linear portion. Solid red line: extrapolation of the linear portion of the curve used to estimate the apparent persistence length. Dotted red lines: 95% confidence interval.
(TIF)

**S1 Video. Cropped cryo-electron tomogram of CHIKV spherules containing the dsRNA at the plasma membrane.** The area corresponds to the tomogram shown in Fig 1A. The video displays consecutive XY planes through the volume.
(AVI)

## Acknowledgments

We would like to thank Felix Bäuerlein (Georg-August-Universität Göttingen) for his input regarding the persistence length calculations, Igor Iashchishyn (Umeå universitet) for his input on the wormlike chain model, and Eliot Rohr for assistance with one script.

## Author Contributions

**Conceptualization:** Timothée Laurent, Lars-Anders Carlson.

**Formal analysis:** Timothée Laurent, Lars-Anders Carlson.

**Investigation:** Timothée Laurent, Lars-Anders Carlson.

**Methodology:** Timothée Laurent, Lars-Anders Carlson.

**Writing – original draft:** Timothée Laurent, Lars-Anders Carlson.

**Writing – review & editing:** Timothée Laurent, Lars-Anders Carlson.

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
