## [Decision Letter · Decision Letter 0]

9 Mar 2023

Dear %TITLE% Carlson,

Thank you very much for submitting your manuscript "The organization of double-stranded RNA in the chikungunya virus replication organelle" for consideration at PLOS Neglected Tropical Diseases. As with all papers reviewed by the journal, your manuscript was reviewed by members of the editorial board and by several independent reviewers. The reviewers appreciated the attention to an important topic. Based on the reviews, we are likely to accept this manuscript for publication, providing that you modify the manuscript according to the review recommendations. 

Sincerely,

Susan M Bueno

Academic Editor

Andrea Marzi

Section Editor

Reviewer's Responses to Questions

**Key Review Criteria Required for Acceptance?**

**Methods**

-Are the objectives of the study clearly articulated with a clear testable hypothesis stated?

-Is the study design appropriate to address the stated objectives?

-Is the population clearly described and appropriate for the hypothesis being tested?

-Is the sample size sufficient to ensure adequate power to address the hypothesis being tested?

-Were correct statistical analysis used to support conclusions?

-Are there concerns about ethical or regulatory requirements being met?

Reviewer #1: 1. The objectives of the study are clear with a testable hypothesis.

2. The study design is appropriated to address the objectives.

3. The population is clearly described.

4. The sample size is sufficient, specially given the technical limitations.

5. The statistical analysis is correct.

6. I do have a concern about the regulatory requirements. This based on the fact that the authors do not describe at all the methodology to induce the assembly of the vesicle in replicon expressing cells. There is no description of how the experiments were performed and in order to go from transfected or infected cells to Cryo-EM. Therefore, the methodology section must be improved.

Reviewer #2: In this research, the authors followed a well-designed methodological strategy to prove their hypothesis with sufficient veracity. The objectives of the work were appropriately described and proper scientific reasoning was followed to demonstrate their proposals about the assembly of the replication spherules of the Chikungunya virus, and its possible use for the design of more efficient RNA vaccines, with which the synthesis of proteins of vaccine interest is achieved for a better sensitization and activation of the immune system. 

Based on the results of the investigation, they make correct statements about their proposal; some significant statements were:

1. The viral protein nsP1 serves as a base for the assembly of a larger protein complex at the neck of the membrane bud; But in the absence of any negatively charged lipids, nsP1 did not bind appreciably to the vesicles; and nsP1 has concentration-dependent binding to PS-containing membranes. 

2. Spherules contain a single copy of the viral genome in double-stranded form.

3. The energy released by RNA polymerization is found to be sufficient to remodel the membrane to the characteristic spherule shape.

4. Alphaviruses are not only a major source of morbidity, but their unique RNA replication mechanism is also used to develop self-replicating RNA vaccines that induce a more potent immune response than conventional mRNA vaccines

Reviewer #3: No major experiments/analyses are required for this study.

**Results**

-Does the analysis presented match the analysis plan?

-Are the results clearly and completely presented?

-Are the figures (Tables, Images) of sufficient quality for clarity?

Reviewer #1: 1. The analysis match the plan.

2.1 The results are not always clearly presented and some discussion seem incomplete. In particular, it is not clear what do the class averages in figure 1C-G correspond in a vesicle like the one on figure 1A. Also, it is not clear if the apparent persistence length depends on the size of the vesicle. Please clarify these problems.

2.2 Interestingly they found a P < 63 nm. However, when they calculate ro zero and correlate that with the phase diagram they use P = 63 nm. I would suggest to repeat this calculation with P = 25-32 nm, I am sure this P will result in a scenario that agrees with your results.

2.3 In the discussion they claim that P = 23 might be an underestimation. However, this might not be true. For example Cifra et al 2008 JPB showed that the persistence length of a worm-like polymer under confinement changes as a function of the confinement under certain stiffness regimes. They predicted that for such polymer you should see a toroidal structure as you propose in figure 3b. The fact that you do not observe that structure might suggest that P is lower than expected because the RNA inside is not 100% ds. In fact, based on Cifra´s work P should increase for a worm-like polymer, hence the RNA inside the spheres might be more complex than expected. I would suggest to take this in consideration to analyze the results.

2.4 it is possible that the dsRNA is bound to proteins that could condense it and thus lower the persistence length? Also, please discuss more the possibility that the RNA is not completely ds.

2.5 At some point the mentioned 80% of the RNA is present as dsRNA, but this is not consistent with table 1. Please clarify.

2.6 The statements in lines 215-218. On the one hand, they are right that there no single conformation. On the other hand, they got the answer wrong; dsDNA is statisical polymers with an ensemble of conformations. Hence, just the nature of dsRNA is enough to not have a single conformation.

Reviewer #2: The results described in the manuscript are relevant and original, since they not only describe the structure and stability of the replication spherules of this virus, which is important to clarify the replication mechanisms of this virus, but also open the possibility of its use as a gene expression strategy of interest in vaccines and other therapeutic strategies.

As mentioned above, the experimental strategy was carefully designed; perhaps in the study of the participation of phospholipids in the binding of NSP1 to the membrane, some experiment of colocalization of NSPs proteins with antibodies could be included to ensure that NSP1 is the anchor for other NSPs. The colocalization study would only serve to confirm how NSP1 participates in the union of the other NSPs in the organization of the spherules and its possible role in viral replication.

Reviewer #3: Most results are clearly represented, and figures are of sufficient quality.

**Conclusions**

-Are the conclusions supported by the data presented?

-Are the limitations of analysis clearly described?

-Do the authors discuss how these data can be helpful to advance our understanding of the topic under study?

-Is public health relevance addressed?

Reviewer #1: 1. the conclusions are partially supported by the data presented. I would suggest to also analyze how the distribution of the RNA changes by repeating the same calculation but with P = 23 nm. Perhaps, this calculations will result in regime that is consistent with the data (it could lower the amount of estimated dsRNA in line 280).

2. The limitations of the method with respect to the measurements are not clearly addresses. This might be because the group is an expert in these techniques but it would benefit the reader to have such discussions.

3. Given the scope of the journal it would be good to discuss how these data can be helpful to advance our understanding of the topic under study and its relevance in public health.

Reviewer #2: The authors show a scientific dialogue with previous studies and are supported by results that they have previously described. It is important to mention that tomography coupled with microscopy has contributed to an advance in the knowledge of this area of research.

The research shows a substantial advance in the knowledge of viral replication spherules and their possible application in the development of vaccines and other therapeutic strategies such as the expression of genes that regulate certain metabolic pathways or silencing RNAs.

Reviewer #3: Discussions and conclusions are supported mainly by the data. However, their contents seem too technical and specific, making it difficult for the general reader to understand.

**Editorial and Data Presentation Modifications?**

Reviewer #1: The editorial and data presentation modifications are minimal. It would be better go have a detailed methodology of the work with cells, how some of the data was treated. Most of my comments are in the previous sections.

Reviewer #2: For all of the aforementioned, I recommend that the manuscript be accepted.

Reviewer #3: N/A

**Summary and General Comments**

Reviewer #1: The strength of this article is the high quality of the data and the careful analysis to understand the physical properties of dsRNA inside the replication vesicles. To the best of my knowledge this is a novel research that could inspire theoretical physicist to re-evaluate classical models. 

I have requested a minor revision because my observations have be easily addressed.

Reviewer #2: No comment

Reviewer #3: In this manuscript, Laurent and Carlson extended their previous Cryo-EM study of the CHIKV spherules and provided some new insights into the structural organization of CHIKV dsRNA in the spherules. Although the spherules the authors analyzed were not the authentic ones formed in CHIKV-infected cells, perhaps future work will pick up this slack through the use of infectious viruses. Particularly, it will be intriguing to know how the 26S subgeneric RNA encoding structural proteins are organized.

The manuscript is well-written and would be acceptable in its current form. However, it can be strengthened by discussing whether different conformations of dsRNA (described in Figures 1 and 3) would be functionally different in terms of the single-stranded RNA synthesis and biology of CHIKV.

PLOS authors have the option to publish the peer review history of their article (what does this mean?). If published, this will include your full peer review and any attached files.

Reviewer #1: Yes: Mauricio Comas-Garcia

Reviewer #2: Yes: J Leopoldo Aguilar-Faisal

Reviewer #3: No

Figure Files:

Data Requirements:

Reproducibility:

References

---

## [Decision Letter · Decision Letter 1]

22 May 2023

Dear %TITLE% Carlson,

We are pleased to inform you that your manuscript 'The organization of double-stranded RNA in the chikungunya virus replication organelle' has been provisionally accepted for publication in PLOS Neglected Tropical Diseases.

Best regards,

Susan M Bueno

Academic Editor

Andrea Marzi

Section Editor

Reviewer's Responses to Questions

**Key Review Criteria Required for Acceptance?**

**Methods**

-Are the objectives of the study clearly articulated with a clear testable hypothesis stated?

-Is the study design appropriate to address the stated objectives?

-Is the population clearly described and appropriate for the hypothesis being tested?

-Is the sample size sufficient to ensure adequate power to address the hypothesis being tested?

-Were correct statistical analysis used to support conclusions?

-Are there concerns about ethical or regulatory requirements being met?

Reviewer #1: The authors have successfully address all the issues and I am happy with the current form of the manuscript to be published

Reviewer #3: (No Response)

**Results**

-Does the analysis presented match the analysis plan?

-Are the results clearly and completely presented?

-Are the figures (Tables, Images) of sufficient quality for clarity?

Reviewer #1: The authors have successfully address all the issues and I am happy with the current form of the manuscript to be published

Reviewer #3: (No Response)

**Conclusions**

-Are the conclusions supported by the data presented?

-Are the limitations of analysis clearly described?

-Do the authors discuss how these data can be helpful to advance our understanding of the topic under study?

-Is public health relevance addressed?

Reviewer #1: The authors have successfully address all the issues and I am happy with the current form of the manuscript to be published

Reviewer #3: (No Response)

**Editorial and Data Presentation Modifications?**

Reviewer #1: The authors have successfully address all the issues and I am happy with the current form of the manuscript to be published

Reviewer #3: (No Response)

**Summary and General Comments**

Reviewer #1: The authors have successfully address all the issues and I am happy with the current form of the manuscript to be published

Reviewer #3: The authors have satisfactorily responded to the reviewer's comments.

PLOS authors have the option to publish the peer review history of their article (what does this mean?). If published, this will include your full peer review and any attached files.

Reviewer #1: **Yes: **Mauricio Comas-Garcia

Reviewer #3: No

---

## [Editor Report · Acceptance letter]

30 Jun 2023

Dear %TITLE% Carlson,

We are delighted to inform you that your manuscript, "The organization of double-stranded RNA in the chikungunya virus replication organelle," has been formally accepted for publication in PLOS Neglected Tropical Diseases.

Best regards,

Shaden Kamhawi

co-Editor-in-Chief

Paul Brindley

co-Editor-in-Chief
